# Hidden Population Estimation with Indirect Inference and Auxiliary Information

Justin Weltz[1]  Eric Laber[1,3]  Alexander Volfovsky[1,2]

[1]Department of Statistical Science, Duke University, Durham, North Carolina, USA
[2]Department of Computer Science, Duke University, Durham, North Carolina, USA
[3]Department of Biostatistics and Bioinformatics, Duke University, Durham, North Carolina, USA

## Abstract

Many populations defined by illegal or stigmatized behavior are difficult to sample using conventional survey methodology. Respondent Driven Sampling (RDS) is a participant referral process frequently employed in this context to collect information. This sampling methodology can be modeled as a stochastic process that explores the graph of a social network, generating a partially observed subgraph between study participants. The methods currently used to impute the missing edges in this subgraph exhibit biased downstream estimation. We leverage auxiliary participant information and concepts from indirect inference to ameliorate these issues and improve estimation of the hidden population size. These advances result in smaller bias and higher precision in the estimation of the study participant arrival rate, the sample subgraph, and the population size. Lastly, we use our method to estimate the number of People Who Inject Drugs (PWID) in the Kohtla-Jarve region of Estonia.

## 1 INTRODUCTION

Valid statistical inference tasks require understanding the data sampling mechanism [Heckathorn, 1997]. Often this means identifying a sampling frame, e.g., an enumeration of units in the population of interest, and sampling from it with a known rule. However, many populations lack a conventional sampling frame because they are characterized by behaviors that are illegal [Frost et al., 2006, Johnston et al., 2010] or stigmatized [Hladik et al., 2012, Kerr et al., 2018]. These "hidden" populations include intravenous drug users [Crawford, 2016], undocumented immigrants [Johnston et al., 2010], and other vulnerable groups.

Respondent Driven Sampling (RDS) is a participant referral process frequently employed by researchers when a sampling frame is unavailable because it preserves the privacy and safety of at-risk populations [Heckathorn, 1997]. For example, RDS was used to study HIV incidence and prevalence among people who inject drugs (PWID) in St. Petersburg, Russia [Crawford, 2016, Crawford et al., 2018b, Heimer and White, 2010]. Here, RDS is leveraged because it is easier to engender trust if study participants recruit their own social contacts.

RDS has been similarly employed to study other hidden populations at risk of HIV and other infectious diseases [Remera et al., 2024, Mapingure et al., 2024, Alinaghi et al., 2024, Barry et al., 2024]. Beyond epidemiological studies, RDS is used for sampling hard-to-reach populations such as migrant workers [Tyldum and Johnston, 2014], street children [Johnston et al., 2010], the unhoused population [Bernard et al., 2018], and ethnic minorities [Mullo et al., 2020]. Lastly, RDS is essential when the relevant subpopulation is highly stigmatized [Stahlman et al., 2016, Arayasirikul et al., 2015, Magno et al., 2022, Shahmanesh et al., 2009].

RDS begins with a small convenience sample of individuals, who are interviewed and asked to recruit other members of the target population with a limited number of incentivized coupons provided by the researchers. When individuals redeem their coupons, they receive an incentive, are enrolled in the study, and are asked to recruit as well. Both access and trust are achieved by incentivizing members of the hidden population to recruit along social connections, thereby verifying the safety of participation. Additionally, anonymity is preserved since only the researchers and a participant's recruiter know an individual's membership status.[1]

This method cannot rely on the prophylactic effects of simple random sampling during estimation because it operates along social connections. We will show how improving estimation of the underlying social network between participants in an RDS, and leveraging commonly collected

---

[1]Various additional layers of protection are possible, such as "coupons" being digital and recruiter information remaining anonymous to recruits.

auxiliary information about participants, can lead to more accurate population size estimates.

The current literature has mainly focused on estimating prevalence of health-related characteristics in the hidden population, e.g., HIV [Montealegre et al., 2013] and syphilis [Frost et al., 2006]. In order to conduct inference under this sampling design, researchers create simple approximate models for RDS recruitment, often treating the implicit social network as a nuisance parameter [Gile, 2011, Volz and Heckathorn, 2008]. In recent years, focus has shifted to uncovering more about this underlying graph [Crawford et al., 2018a, Verdery et al., 2017] for its use in downstream estimation. Population size is one such downstream target.

Estimating population sizes is often imperative for assessing the scope of public health crises [Crawford et al., 2018b, Wu et al., 2017]. There are various approaches to estimating the overall size of a hidden population that do not account for the sampling mechanism, such as RDS, adequately and hence may perform poorly. Simple capture-recapture methods require random sampling and so ignore the mechanism altogether [White, 1982], and multiplier methods [Fearon et al., 2017] depend on every survey participant accurately reporting the hidden population membership status of their acquaintances, which is unrealistic in many sensitive contexts. Successive Sampling has been used to estimate population size from RDS samples [Johnston et al., 2010, Gamble et al., 2023], however this method does not incorporate all the network information available. The key problem with these approaches is that they effectively ignore the underlying graph structure in the population. Crawford et al. [2018b] addresses this by estimating the unobserved edges in a subgraph of the population to develop a model for the hidden population size. Estimating missing graph information requires working with a model over a complex combinatorial space, and we illustrate that the proposed maximum likelihood and Bayesian estimators are necessarily biased or sensitive to the specification of the prior [Crawford et al., 2018b].

We make the following three contributions:

1. Debias existing estimators of the underlying social network in an RDS sample via indirect inference. We provide empirical validation of the theoretical performance suggested by our Proposition 1.
2. Develop a two-stage procedure that incorporates commonly collected auxiliary information into the estimation of the social network, providing improvements to hidden population size estimates.
3. When the underlying social network depends on group structure, we derive improved population size estimators. We provide empirical evidence for the robustness of the proposed approaches as compared to state of the art methods.

The rest of the paper is organized as follows: Section 2 pro-

vides background information on the structure of the RDS stochastic process model and its likelihood. In Section 3, we extend the original indirect inference estimator (IIE) of Jiang and Turnbull [2004] to the case of RDS and show that this estimator is less biased than the MLE asymptotically. Section 4 reviews the existing population size estimation approaches, extends them to the case of more general underlying graph structures, and develops a method for incorporating auxiliary information (such as group membership) into the population size estimation procedure. Section 5 and 6 demonstrate, through simulation studies and a case study respectively, the impact of indirect inference estimation and auxiliary information on population size estimation.

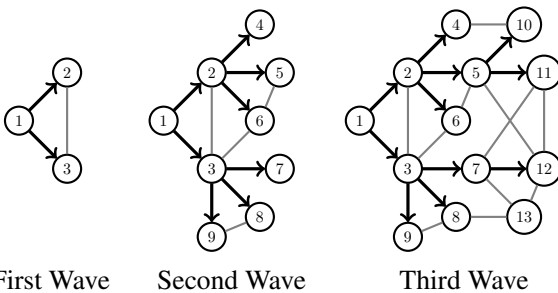

First Wave    Second Wave    Third Wave

Figure 1: $G_R$ is composed of coupon exchanges $\rightarrow$. $G_S$ includes both observed $\rightarrow$ and unobserved connections ——.

## 2 RDS MODEL SETUP AND ISSUES

Throughout we consider a setting where our population is represented by a graph $G = (V, E)$, where $V$ is the set of $|V| = N$ nodes in the graph and $E$ is the set of pairwise connections, or edges, between individuals. Respondent Driven Sampling (RDS) starts with a set of seeds (node 1 in Figure 1), and then proceeds by recruiting other participants (the middle and right panels of Figure 1) over the edges of the original graph $G$. This process continues until a stopping rule is reached (e.g., a predetermined number of recruited individuals or a budget constraint are met). At the end of this process, a researcher is in possession of a recruitment subgraph $G^R = (V^R, E^R) \subset G$ on $|V^R| = n \leq N$ individuals. The labels of the nodes in $V^R$ denote the order in which they arrived at the study (and so participant $i$ was interviewed before participant $j$ if $i < j$). Importantly, this is *not* the vertex induced subgraph of $G$ that would have been observed by projecting the original graph $G$ onto the vertices $V^R$. We will call this induced subgraph $G^S = (V^S, E^S)$ and note that, while $V^S = V^R$, we only know that $E^R \subseteq E^S$. *If we had access to $G^S$ then estimating the size of the graph $G$ would be a simple task.*

There are two reasons that edges in $G^S$ are missing in $G^R$. First, recruiters may run out of coupons before they recruit all of their neighbors (e.g., paritcipant 13 in Figure 1). Second, if participant $i$ recruits participant $k$ before participant

$j$ does, a connection $\{j, k\} \in E^S$ will not be observed because an individual cannot participate in the study multiple times (e.g., participant 6 is recruited by participant 2 before participant 3 can recruit them in Figure 1).

While $G^S$ cannot be observed directly, it can be estimated from data collected during RDS.[2] Typical RDS studies ask participants how many hidden population members they know. For participant $i$, this is their degree in the larger graph $G$, $d_i = |\{\{i, j\} \in E : i \in V^R, j \in V, i \neq j\}|$. The vector of observed degrees, $\mathbf{d} = (d_1, d_2, \ldots, d_n)$, is ordered by arrival to the study. Additionally, we define a vector $\mathbf{w}$ such that $w_i$ is the time between the arrival of participant $i - 1$ and participant $i$. This makes the full data observed at the end of an RDS study $\mathbf{Y} = (G^R, \mathbf{d}, \mathbf{w})$, $\mathbf{Y} \in \mathcal{Y}$.

Our RDS arrival process model is described by wait times attached to edges in $G$ between recruiters with unused coupons and unrecruited members of the hidden population, termed "susceptible edges" [Crawford et al., 2018b]. When the wait time associated with edge $\{i, j\}$ expires, participant $i$ recruits participant $j$ (as long as $j$ has not been previously recruited); $d_j$ and $w_j$ are then recorded and $\{i, j\}$ is added to $G^R$. We assume that edge times are independent and identically distributed according to an exponential distribution [Crawford, 2016].

**Assumption 1 (Exponential Wait Times)** *Upon entering the study, a participant immediately becomes active, and their susceptible edges are assigned a wait time that is drawn independently from an exponential distribution with common parameter $\lambda \in \mathbb{R}^+$. (This combines assumptions 4 and 6 in Crawford et al. [2018b].)*

**Remark 1** *Assumption 1 is common when studying arrival data. It implies Markovian dynamics and leads to a closed form likelihood for the RDS process (Equation 1). It is possible to relax this assumption, e.g., by considering dependence in arrival times. This will lead to changes in the likelihood, but does not preclude the analytic approach we propose.*

Let $A^S \in \{0, 1\}^{n \times n}$ be the adjacency matrix associated with graph $G^S$, where $A^S_{i,j} = 1$ if $\{i, j\} \in E^S$ and 0 if not; let $u_i$ be the number of connections study participant $i$ has to unrecruited hidden population members, $u_i = |\{\{i, j\} \in E : j \notin V^R\}|$, and $\mathbf{u} = (u_1, u_2, \ldots, u_n)$; and let $M$ be the seed set. Additionally, let $\mathrm{lt} : \mathbb{R}^{n \times n} \to \mathbb{R}^{n \times n}$ be the lower-triangular function, i.e., for any $A \in \mathbb{R}^{n \times n}$, we have $\{\mathrm{lt}(A)\}_{i,j} = A_{i,j} 1(i \leq j)$. The joint likelihood for parameters $A^S$ and $\lambda$ is

$$\mathcal{L}_n(\mathbf{Y}|A^S, \lambda) = \left( \prod_{j \notin M} \lambda s_j \right) \exp\left(-\lambda \mathbf{s}^\top \mathbf{w}\right), \quad (1)$$

[2] We use the notation of Crawford et al. [2018b] when possible for referential convenience.

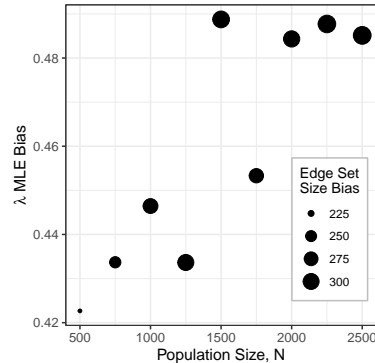

Figure 2: This figure depicts the bias of $\widehat{\lambda}_n$ and $\left|\widehat{E}_n^S\right|$. We can see that the bias of $\lambda$ and the edge set size are positively correlated and increase as the sample proportion decreases.

where $\mathbf{s} = \mathrm{lt}(A^S C)^\top 1 + C^\top \mathbf{u}$ is the susceptible edge vector, and $C \in \mathbb{R}^{n \times n}$ is the coupon matrix in which $C_{ij} = 1$ if participant $i$ has at least one coupon before the $j^{th}$ participant is recruited, and zero otherwise (Definition 4 from Crawford et al. [2018b]). The $i^{th}$ entry of the susceptible edge vector, $s_i \in \mathbf{s}$, is the number of edges between recruiters with coupons and unrecruited members of the hidden population just before the $i^{th}$ study participant is recruited.

Both $G^R$ and $\mathbf{d}$ function as graphical constraints ensuring that the estimated adjacency matrix is *compatible* with the observed data.

**Definition 1 (Compatibility)** *An estimated subgraph $\widehat{G}^S = (\widehat{V}^S, \widehat{E}^S)$ represented by the estimated adjacency matrix $\widehat{A}_n^S$ is compatible with the observed data, $\mathbf{Y}$, if the following three conditions hold: 1. $V^R = \widehat{V}^S$; 2. $E^R \subseteq \widehat{E}^S$; 3. The degree of each $i \in \widehat{V}^S$ does not exceed $d_i$. (This is Definition 5 from Crawford et al. [2018b].)*

Let $\mathcal{A}$ be the space of compatible subgraphs, then the maximum likelihood estimator (MLE) corresponding to Equation (1) is

$$\left\{\widehat{A}_n^S, \widehat{\lambda}_n\right\} = \arg \max_{A^S \in \mathcal{A}, \lambda \in \mathbb{R}^+} \mathcal{L}_n(\mathbf{Y}|A^S, \lambda). \quad (2)$$

## 2.1 ISSUES WITH MAXIMUM LIKELIHOOD ESTIMATION

Beyond computational difficulties associated with maximizing functions over graph space, the MLE in Equation (2) can exhibit severe bias even for moderately large sample sizes. We start by noting that *if* $A^S$ were known, Equation (1) reduces to the likelihood of exponentially distributed data. It is well known that the MLE for the rate parameter of an exponential, $\lambda$, has a bias that diminishes as the sample

size, $n$, increases: $|\mathbb{E}(\widehat{\lambda}_n) - \lambda| = \lambda/(n-1)$. However, in RDS, $A^S$ is not known, *and* the magnitude of the bias is related to the rate of increase of both $n$ and $N$ (the unobserved population size). Specifically, when $A^S$ is unknown, Equation (1) has $n+1$ unknown parameters that are meant to be estimated based on $n$ observations and the graphical constraints imposed by $G^R$ and $\mathbf{d}$ — while the parameters remain identifiable due to these constraints, high quality estimation may not be possible. This is especially true for RDS, as the constraints are often loose in this context ($n \ll N$ and so $n/N \nrightarrow 1$).

In Figure 2, we plot the observed absolute biases in $|\widehat{E}_n^S|$ and $\widehat{\lambda}_n$ following an RDS simulated according to the generative model in Equation (1) with $\lambda = 1$, a single seed participant, five coupons per participant, and $n = 100$. The population graph, $G$, is simulated from an Erdos-Renyi model with edge probability $p$ set to keep the expected degree 10 (details about the Erdos-Renyi model are provided in Section 4). On the x-axis, we vary the *total* population size, $N$. We see that as $n/N$ decreases and the constraints loosen, the bias increases. The intuition behind this is as follows. For a given $\lambda$ and $i \in \{1, 2, \ldots, n\}$, the MLE of $s_i$ *without* graphical constraints is $1/(\lambda w_i)$, which has expectation $\mathbb{E}\{1/(\lambda w_i)\} = \infty$. This suggests that if $n/N \nrightarrow 1$ as $n \to \infty$ and $N \to \infty$, then the MLE of $s_i \lambda$ will have positive bias. RDS is used in settings where $n << N$ (and so the constraints on $\mathbf{s}$ are minimal), so an alternative to the MLE is needed for high quality inference. *We aim to resolve these biases using an alternative estimator motivated by concepts from indirect inference.*

# 3 INDIRECT INFERENCE ESTIMATOR

We define the indirect inference estimator, derive its theoretical properties, and demonstrate its improvement in estimating RDS model parameters empirically.

## 3.1 INDIRECT INFERENCE

The indirect inference estimator (IIE) relies on specifying a calibration statistic. The choice of this statistic is not unique, but often there is a natural option in a given problem domain [Jiang and Turnbull, 2004]; we use the MLE for $\lambda$ as our calibration statistic. The IIE is constructed by finding parameter settings under which the expected value of the calibration statistic matches its observed value.

To formalize the indirect inference estimator (IIE) in our setting, we require a few definitions. Let $\lambda^\dagger : \mathcal{Y} \to \mathbb{R}$ and $A_\dagger^S : \mathcal{Y} \to \{0, 1\}^{n \times n}$ be functions that map the data, $\mathbf{Y}$, to the solutions of Equation (2). Additionally, define $A_\lambda^S : \mathcal{Y} \times \mathbb{R}^+ \to \{0, 1\}^{n \times n}$ so that for observed data, $\mathbf{Y}$, and value $\lambda' > 0$, $A_\lambda^S(\mathbf{Y}, \lambda')$ is the solution to Equation (2) holding $\lambda$ fixed at $\lambda'$.

We propose the following estimation procedure for our model parameters. Let $\widetilde{\lambda}_n$ solve

$$\mathbb{E}_{\mathbf{Z} \sim P_{A_\lambda^S(\mathbf{Y}, \widetilde{\lambda}_n), \widetilde{\lambda}_n}} \left\{ \lambda^\dagger(\mathbf{Z}) \right\} = \lambda^\dagger(\mathbf{Y}), \qquad (3)$$

and $\widetilde{A}_n^S = A_\lambda^S(\mathbf{Y}, \widetilde{\lambda}_n)$, then the IIE is the pair $(\widetilde{\lambda}_n, \widetilde{A}_n^S)$. The expectation in Equation (3) is taken over simulated data $\mathbf{Z} = (G^R, \mathbf{d}, \mathbf{w}^*) \in \mathcal{Y}$, where $\mathbf{w}^* \sim P_{A^S, \lambda}$ and $P_{A^S, \lambda}$ is the generative model described in Equation (1). The procedure for calculating the IIE is summarized in Algorithm 1. This algorithm requires $K \times J$ evaluations of the MLE. Because these evaluations are embarrassingly parallelizable, the IIE has the same computational complexity as the MLE.

To understand *why* an IIE can reduce bias, we first discuss the IIE for exponentially distributed data, which we observed in Section 2.1 are closely related to the data generated by RDS. The important benefit of this setting is that we are able to derive the analytic form of the IIE.

Suppose $\mathbf{X} = (X_1, ..., X_n)$ comprises $n$ independent draws from an exponential distribution indexed by $\lambda \in \mathbb{R}^+$. The likelihood of $\mathbf{X}$ is

$$\mathcal{L}(\lambda | \mathbf{X}) = \prod_{i=1}^{n} \lambda \exp(-\lambda X_i) = \lambda^n \exp(-\lambda \sum_{i=1}^{n} X_i).$$

The MLE is $\widehat{\lambda}_n = n/(\sum_{i=1}^n X_i)$, which is distributed according to an Inverse-Gamma distribution with shape and scale parameters $(n, n\lambda)$. The absolute bias of the MLE is $|\mathbb{E}(\widehat{\lambda}_n - \lambda)| = \lambda/(n-1)$. Choosing the MLE as the calibration statistic in the IIE procedure, we see that the IIE is $\widetilde{\lambda}_n = (n-1)/(\sum_i X_i)$, which is unbiased.

Moreover, the mean squared error (MSE) of $\widetilde{\lambda}_n$ is smaller than that of $\widehat{\lambda}_n$:

$$\mathrm{MSE}(\widehat{\lambda}_n) = \frac{\lambda^2(n^2 + n - 2)}{(n-1)^2(n-2)},$$

$$\mathrm{MSE}(\widetilde{\lambda}_n) = \frac{\lambda^2}{(n-2)},$$

$$\mathrm{MSE}(\widehat{\lambda}_n) - \mathrm{MSE}(\widetilde{\lambda}_n) = \frac{3\lambda^2(n-1)}{(n-1)^2(n-2)} > 0.$$

In general, the IIE (that uses the MLE as a calibration statistic) is unbiased for a parameter if the bias of the MLE is linear in the parameter. The exponential likelihood example above suggests that this is possible in our setting. We formally compare the asymptotic behavior of the IIE bias to the MLE bias in the next subsection.

## 3.2 ASYMPTOTICS

To characterize the asymptotic behavior of the IIE, we assume that the MLE admits an Edgeworth expansion [Hall, 2013].

**Algorithm 1:** The Indirect Inference Estimator

**Goal:** Find the estimator,

$$\widetilde{\lambda}_n \in \arg\min_{\lambda} \left| \mathbb{E}_{\mathbf{Z} \sim P_{A_\lambda^S(\mathbf{Y}, \lambda), \lambda}} \left\{ \lambda^\dagger(\mathbf{Z}) \right\} - \lambda^\dagger(\mathbf{Y}) \right|$$

Generate a grid of $\lambda^k$ values, $k \in \{1, 2, .., K\}$, centered at $\widehat{\lambda}_n$

**for** *k in* $\{1, 2, \ldots, K\}$ **do**

    **for** *j in* $\{1, 2, \ldots, J\}$ **do**

        Find $\widehat{A}_{n,k}^S = A_\lambda^S(\mathbf{Y}, \lambda^k)$

        Simulate wait time vector $w^{k,j}$ from the model defined by parameters $\widehat{A}_{n,k}^S, \lambda^k$

        Find $\widehat{\lambda}_n^{k,j}$ by solving Equation (2) with generated data $\mathbf{Z}^{k,j} = (G^R, \mathbf{d}, w^{k,j})$

    **end**

    Save set $\left\{ \lambda^k, \widehat{A}_{n,k}^S, \widehat{\lambda}_n^k = \left( \sum_{j=1}^J \widehat{\lambda}_n^{k,j} \right) / J \right\}$

**end**

Calculate $k^* = \arg\min_{k \in \{1, 2, \ldots, K\}} \left| \widehat{\lambda}_n^k - \lambda^\dagger(\mathbf{Y}) \right|$

Output estimators $\left\{ \widetilde{\lambda}_n, \widetilde{A}_n^S \right\} = \left\{ \lambda^{k^*}, \widehat{A}_{n,k^*}^S \right\}$

---

**Assumption 2** *As* $n \to \infty$,

$$\widehat{\lambda}_n = \lambda + \frac{A(V, \lambda)}{\sqrt{n}} + \frac{B(V, \lambda)}{n} + \frac{C(V, \lambda)}{n^{3/2}} + o_p(n^{-3/2}),$$

*where V has a distribution that does not depend on λ, and* $A(V, \lambda)$ $B(V, \lambda)$ *and* $C(V, \lambda)$ *are random vectors that only depend on λ and V.*

Such an expansion holds for the MLE under general conditions; see Section 2.4 of Hall [2013] for details. Under this expansion, it can be seen that the bias is of order $n^{-1/2}$.

**Proposition 1** *Given Assumption 2, as* $n \to \infty$,

$$\mathbb{E}\left( \widetilde{\lambda}_n \right) = \lambda + \frac{\mathbb{E}\left\{ C^*(V, \lambda) \right\}}{n^{3/2}} + o_p(n^{-3/2}),$$

*where V is a random variable with a distribution that does not depend on λ, and* $C^*(V, \lambda)$ *is a random vector that only depends on λ and V.*

Proposition 1 follows from Assumption 2 and Corollary 2.1 in Gouriéroux et al. [2000]. It shows that the IIE does not have bias terms of orders $n^{-1/2}, n^{-1}$, while the MLE does.

## 3.3 EMPIRICAL PERFORMANCE: STUDY PARTICIPANT ARRIVAL RATE AND SUBGRAPH ACCURACY IMPROVEMENTS

In this section, we empirically evaluate the finite sample behavior of our proposed IIE estimator for the two model

Table 1: Graph True Positive Rate (%)

| Pop. | Deg. | MLE | | IIE | |
|---|---|---|---|---|---|
| | | Average | Std. | Average | Std. |
| 1000 | 5 | 56.66 | 0.85 | 67.61 | 1.47 |
| 1000 | 10 | 36.52 | 0.82 | 50.48 | 2.00 |
| 1000 | 15 | 29.49 | 0.79 | 47.71 | 2.38 |
| 5000 | 5 | 58.76 | 0.96 | 69.93 | 1.58 |
| 5000 | 10 | 37.00 | 0.91 | 51.73 | 1.92 |
| 5000 | 15 | 30.57 | 1.08 | 49.48 | 2.48 |
| 10000 | 5 | 59.25 | 0.93 | 72.18 | 1.52 |
| 10000 | 10 | 37.52 | 0.93 | 54.15 | 2.08 |
| 10000 | 15 | 30.50 | 0.84 | 51.30 | 2.22 |

These are the true positive rates of the estimated subgraphs over a series of population sizes (Pop.) and average degrees (Deg.). The standard deviations reported quantify the Monte Carlo error associated with these estimates based on 100 simulations.

parameters in the likelihood of Equation (1). We simulate RDS trajectories of size 100 over various graph sizes, with an average wait time of $1/\lambda = 1$ and each recruit having 5 coupons. The hidden population graph, $G$, is simulated from an Erdos-Renyi model with edge probability $p$ (details of this model choice are provided in Section 4). In our simulations, we vary $N \in \{1000, 5000, 10000\}$ and $Np \in \{5, 10, 15\}$. Algorithm 1 is used to construct the IIE, which we compare to the MLE.

Table 1 demonstrates that the IIE for the sample subgraph, $\widetilde{A}_n^S$, has a higher true positive rate than the MLE in all simulation settings. Importantly, Table 3 in Appendix A shows that these improvements do not come at the expense of the true negative rate.

The rate parameter $\lambda$ is of independent interest for assessing coupon uptake speed and the time necessary for recruiting a target sample size. Table 4 in Appendix A indicates that over a range of population sizes and graph densities, the IIE, $\widetilde{\lambda}_n$, outperforms the MLE in terms of MSE.

**Remark 2** *Consistent with Figure 2 and the intuition developed in Section 2.1, the advantage of both* $\widetilde{\lambda}_n$ *and* $\widetilde{A}_n^S$ *over* $\widehat{\lambda}_n$ *and* $\widehat{A}_n^S$ *respectively is slightly greater in high average degree and low sample proportion settings generally.*

## 4 HIDDEN POPULATION SIZE ESTIMATION

One of the primary goals of sampling hidden populations is to estimate their total size, $N$. Imagine that the population graph, $G = (V, E)$, is a sample from an Erdos-Renyi graph model with parameters $N$ and $p$ (that is, there are $N$

individuals in the graph and the probability of a connection between any two of them is $p$). While this is a very simple model, it has demonstrated practical utility when estimating hidden population size, forming the basis for methods such as the snowball sampling estimator [Frank and Snijders, 1994] and the network scale-up estimator [Killworth et al., 1998]. Under an Erdos-Renyi model, the degree of each individual in $G$ is distributed as $d_i \sim \text{Binomial}(N-1, p)$. If we had access to a simple random sample of individuals, then we could directly estimate $N$ based on this likelihood.

As discussed earlier, RDS does not yield a simple random sample from the population (e.g., an individual's probability of being sampled depends on their degree [Heckathorn, 1997, Gile, 2011]). Conditional on the (unobserved) $A^S$ and the Erdos-Renyi assumption, it is possible to write down the distribution for the number of edges individual $i \in \{1, 2, \ldots, n\}$ shares with unsampled members of the hidden population at the time of individual $i$'s recruitment. Let $d_i^u = d_i - \sum_{j=1}^{i-1} \mathbb{I}\left(\{i, j\} \in E^S\right)$, and note that, unlike $d_i$, this quantity is independently and identically distributed from a Binomial distribution, $d_i^u \sim \text{Binomial}(N-i, p)$.

Sections 4.1 and 4.2 present population size estimators based on an Erdos-Renyi graph assumption. This is for simplicity of exposition since there are only two parameters in this model, $N$ and $p$. However, the proposed approach can easily be applied to other graph models by deriving the corresponding distribution of $d_i^u$. For example, consider a population graph that is distributed according to a stochastic block model (SBM) with two groups, $V_A \subseteq V$ and $V_B = V \setminus V_A$. Let the probability of an edge between members of the same group be $p_{\text{in}}$, and the probability of a connection between members of different groups be $p_{\text{out}}$. Assume that we observe the group membership of each study participant, and define $i_A, i_B$ as the number of individuals in groups $A$ and $B$ respectively that are recruited before participant $i+1$.

$$\mathbb{P}(d_i^u = d) = \sum_{j=0}^{d} \binom{N_i}{j} \binom{(N-i) - N_i}{d - j} p_{\text{in}}^j (1 - p_{\text{in}})^{N_i - j}$$
$$\times p_{\text{out}}^{d-j} (1 - p_{\text{out}})^{(N-i) - N_i - (d-j)},$$

where $N_A \geq N_B$ and $p_{\text{in}} \geq p_{\text{out}}$. We can use this distribution to estimate $N$ in the SBM setting. Appendix B provides details of those derivations.

## 4.1 REVISING CURRENT APPROACHES

Based on this population size model, Crawford et al. [2018b] propose an approximate Bayesian MCMC sampling scheme with strong priors on $p$ and $A^S$ to conduct inference on $N$. They find that informative priors on $p$ are necessary for ensuring finite first and second moments of the posterior distribution for $N$. For example, the most diffuse prior on $p$ in their simulations has a variance of about $5 \times 10^{-6}$.

Moreover, they use an informative prior on the graph space $\pi(A^S) \propto \exp(-\gamma|E^S|)$, where $\gamma = -\log\{p/(1-p)\}$ ranges from about 5 to 9, imposing heavy penalties on graphs with large edge sets. These priors inflate the posterior mean of $N$, resulting in significant upward bias.

Prior selection is non-trivial in our problem. Choosing a non-informative prior risks an improper posterior [Kahn, 1987], but, given the nature of the populations we aim to study, it is unlikely that strong informative priors are scientifically justifiable. Moreover, full posterior inference for $N$ is not possible due to computational constraints, requiring multiple approximations [Hunter and Handcock, 2006, Crawford et al., 2018b]. We avoid these issues by reformulating the problem as regularized estimation, which incorporates information on edge prevalence, $p$, via a regularization term. Given regularization function $R(\check{p}) = \log \text{Beta}(\check{p}; a, b)$ for $a, b \in \mathbb{R}^+$, we define the regularized estimates of $N, p$ conditional on $\widehat{A}_n^S$ and $\widetilde{A}_n^S$ as

$$\left\{\widehat{p}, \widehat{N}\right\} = \arg\max_{\check{p}, \check{N}} \log \mathcal{L}(\check{N}, \check{p} | \widehat{A}_n^S) + R(\check{p}),$$

$$\left\{\widetilde{p}, \widetilde{N}\right\} = \arg\max_{\check{p}, \check{N}} \log \mathcal{L}(\check{N}, \check{p} | \widetilde{A}_n^S) + R(\check{p}).$$

## 4.2 IMPROVING ESTIMATION USING AUXILIARY INFORMATION

The RDS data collection process commonly includes a large survey that can be used to improve population size estimation [e.g. Frost et al., 2006, Wu et al., 2017]. In particular, it is common to track how information accumulates over the RDS process, and this measurement necessarily carries information about the underlying network. For example, an RDS interview may begin with a quiz about local free resources, important public health issues, or beneficial health practices (e.g., for People Who Inject Drugs this might include drug therapy options or needle exchange sites). The interview ends with the interviewer revealing the answers to the quiz so that each study participant leaves the study with the same amount of information. The performance of a study participant on this quiz is a graph dependent outcome, **Q**. Below we propose a model for **Q** that, when combined with the IIE approach of Section 3, substantially improves the population size estimates of the previous section.

**Remark 3** *Other graph-dependent outcomes are certainly possible: measurements may depend on participant interactions with their friends or require participants to quantify some characteristic of their referral chain. These different types of* **Q** *would simply require different models from the ones we study below, but could otherwise be easily incorporated into the analysis.*

Define monotonically increasing functions $f : \mathbb{R} \to \mathbb{R}$ and $g : \mathbb{R} \to \mathbb{R}$, $\mathbf{1}_n = (1, 1, \ldots, 1) \in \mathbb{R}^n$, and zero-mean

distribution $F$. If we assume that there is communication over the network, then the performance of an interviewee on the quiz should be proportional to their connections to previously recruited study participants,

$$q_i = f\{\alpha + \gamma g(m_i)\} + \epsilon_i, \ \epsilon_i \overset{\text{i.i.d}}{\sim} F, \qquad (4)$$

where $\mathbf{m} = \{A^S \cdot \mathrm{lt}(\mathbf{1}_n \mathbf{1}_n^\top)\} \mathbf{1}_n$. The $i^{th}$ entry of $\mathbf{m}$ is the number of neighbors of study participant $i$ who were recruited before participant $i$. In Equation (4), $\alpha$ represents an average hidden population member's knowledge of the quiz subject without outside intervention and $\gamma$ is the intensity of communication flow. Adding information about $\mathbf{Q} = (q_1, q_2, \ldots, q_n)$ to our analysis will improve estimation of $\mathbf{m}$, which will improve estimators of $A^S$ and $N$.

We now augment our IIE procedure with the auxiliary information contained in $\mathbf{Q}$. We expand $\mathbf{Y}$ to include the regression information, $\mathbf{Y}^r = (\mathbf{Q}, G^R, d, \mathbf{w}) \in \mathcal{Y}^r$. Define $\lambda_r^\dagger : \mathcal{Y}^r \to \mathbb{R}$ as the function that maps the data, $\mathbf{Y}^r$, to the MLE for $\lambda$. Additionally, define $A_\lambda^{S,r} : \mathcal{Y}^r \times \mathbb{R} \to \{0,1\}^{n \times n}$ so that for value $\lambda' > 0$, $A_\lambda^{S,r}(\mathbf{Y}^r, \lambda')$ is the MLE estimator of $A^S$ holding $\lambda$ fixed at $\lambda'$. Let $\widetilde{\lambda}_n^r$ solve

$$\mathbb{E}_{\mathbf{Z}^r \sim P_{A_\lambda^{S,r}(\mathbf{Y}^r, \widetilde{\lambda}_n^r), \widetilde{\lambda}_n^r}} \left\{ \lambda_r^\dagger(\mathbf{Z}^r) \right\} = \lambda_r^\dagger(\mathbf{Y}^r), \qquad (5)$$

and $\widetilde{A}_n^{S,r} = A_\lambda^{S,r}(\mathbf{Y}^r, \widetilde{\lambda}_n^r)$, then the IIE is now the pair $(\widetilde{\lambda}_n^r, \widetilde{A}_n^{S,r})$. The expectation in Equation (5) is over simulated data $\mathbf{Z}^r = (\mathbf{Q}, G^R, \mathbf{d}, \mathbf{w}^*) \in \mathcal{Y}$, where $\mathbf{w}^* \sim P_{A^S,\lambda}$ and $P_{A^S,\lambda}$ is the generative model described in Equation (1). Algorithm 2 in Appendix E builds on Algorithm 1 and provides the complete description for this computation. The regularized estimator of population size conditional on the IIE with auxiliary information is

$$\left\{ \widetilde{p}^r, \widetilde{N}^r \right\} = \arg\max_{\check{p}, \check{N}} \log \mathcal{L}(\check{N}, \check{p} | \widetilde{A}_n^{S,r}) + R(\check{p}).$$

# 5 POPULATION SIZE ESTIMATION SIMULATIONS

In this section, we empirically evaluate the IIEs of hidden population size with and without auxiliary information. The first simulation study compares our estimators to state-of-the-art competitors for a variety of population sizes and graph densities. The second and third simulations showcase the robustness of our estimators to different graph models.

**Simulation 1.** For each simulation, we draw a hidden population graph from an Erdos-Renyi model, $G \sim \mathrm{ER}(N, p)$, varying $N \in \{1000, 5000, 10000\}$ and $Np \in \{5, 10, 15\}$. We then simulate an RDS study of size $n = 100$ over this graph, starting from 3 random seeds. This RDS follows the generative model specified in Equation (1) with $\lambda = 1$ and 5 coupons. Letting $\mathbf{m} = \{A^S \cdot \mathrm{lt}(\mathbf{1}_n \mathbf{1}_n^\top)\} \mathbf{1}_n$, we observe a

vector of study participant attributes, $\mathbf{Q} = (q_1, q_2, \ldots, q_n)$, drawn according to

$$q_i = \alpha + \gamma m_i + \epsilon_i, \quad \epsilon_i \overset{\text{i.i.d}}{\sim} N(0, \sigma^2),$$

which is within the class of models outlined in Equation (4). We set $\alpha = 0$, $\gamma = 1$, and $\sigma^2 = 1$ and experiment with regularization information on $p$ to explore the utility of social network edge density information when estimating population size. This procedure is repeated 200 times for each simulation setting.

We compare our estimators to the MLE derived in Crawford et al. [2018b] as well as to several estimators proposed in Handcock et al. [2014] that use the successive sampling (SS) method. Under a uniform prior, the SS estimators, which are posterior summaries, require the researcher to specify the maximum that the population size can attain, $N_{\max}$. For a given $N$, we use values $N_{\max} \in \{3N, 5N, 8N\}$.

Figure 3 reports the results across all nine simulation setups. First, we note that the IIEs with and without auxiliary information have lower maximum absolute deviation (MAD) than the MLE over a range of hidden population graph sizes and densities. The weak regularization information setting is defined by $R(\check{p}) = \log \mathrm{Beta}(\check{p}; a, b)$, where $\mathrm{Beta}(\check{p}; a, b)$ is centered at $p$ with $a = 0.1$. Consistent with Remark 2, the improvements of the IIE without auxiliary information over the MLE are greater in high average degree settings. The improvements of the IIE with auxiliary information over the IIE without auxiliary information follow the same pattern. When comparing to the SS approach, we note that the estimators based on this procedure are very sensitive to the prior specification. In fact, Figure 3 shows that the MAD for the SS Mean estimator (the posterior mean) where $N_{\max} = kN$ for $k \in \{3, 5, 8\}$ is almost exactly $|(k-2)N - n|/2$, which is the absolute difference between the prior mean and $N$.

In Appendix D, we explore the role of regularization in our estimator. Figure 6 in Appendix D shows that in the strong regularization setting, where $R(\check{p}) = \log \mathrm{Beta}(\check{p}; a, b)$ and $a = 10$, the improvements of $\widetilde{N}$ and $\widetilde{N}^r$ over $\widehat{N}$ are greater in larger populations.

**Simulation 2: Stochastic Block Model.** We assess the sensitivity of our population size estimate results to the Erdos-Renyi model assumption. Following Crawford et al. [2018b] and Gile et al. [2018], we divide the hidden population into two groups, $V_A \subseteq V$ and $V_B = V \setminus V_A$. The probability of an edge between members of the same group is $p_{\mathrm{in}}$, and the probability of a connection between members of different groups is $p_{\mathrm{out}}$. For constant $c \in [0, 1]$, we set $p_{\mathrm{out}} = cp_{\mathrm{in}}$. Lastly, we let $p^* = \mathbb{P}(\{i, j\} \in E)$, where nodes $i$ and $j$ are drawn uniformly at random from $V$.

In this simulation, we set $N = 5000$, $p^* = 0.002$ (implying an average degree of 10), $c = 0.3$ and $N_A/N = 0.6$. We report estimates assuming an underlying SBM with strong

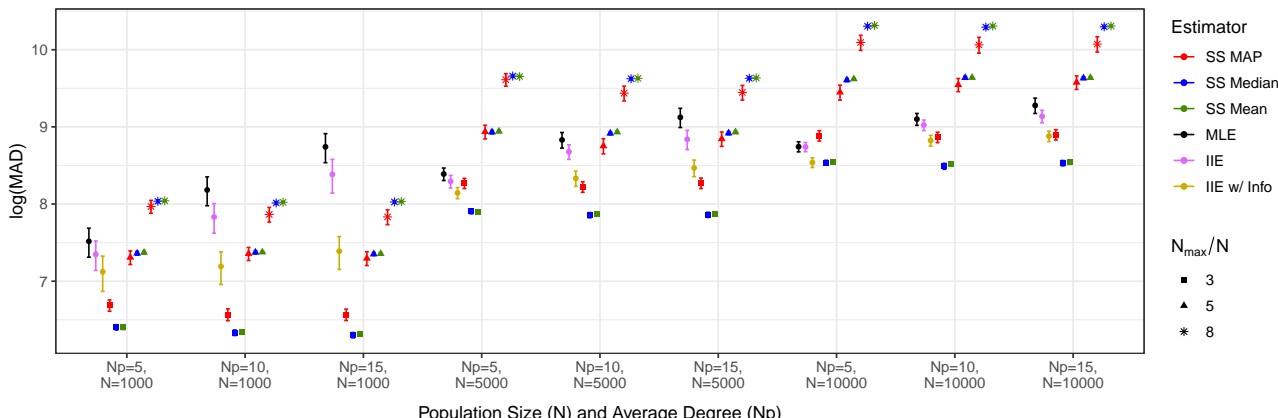

Figure 3: This figure compare the performance of successive sampling estimators, $\widehat{N}$, $\widetilde{N}$, and $\widetilde{N}^r$ under weak regularization information over a series of population sizes, $N$, and average degrees, $Np$, with 90% Monte Carlo confidence intervals.

regularization around $p_{\text{in}}$ and $p_{\text{out}}$ (details in Appendix B) and estimates under misspecification of the network model as an Erdos-Renyi. Figure 4 in Appendix B indicates that the estimator error when the network model is correctly specified follows the same pattern as Figure 3. Figure 4 also shows that ignoring the block structure results in significantly worse estimator performance. Additional analysis under network model misspecification is in Appendix B. Encouragingly, both the correctly and incorrectly specified IIEs perform better than their MLE counterparts.

**Simulation 3: Latent Space Model.** We further assess the sensitivity of our estimation techniques by generating network data from the more general latent space inner product model. We allow edge probabilities to range from $1.5 \times 10^{-3}$ to $2.7 \times 10^{-3}$ with an expected degree of 10. In this context, we use estimators that assume the Erdos-Renyi model. Figure 5 in Appendix C indicates that these estimators, while being incorrectly specified, still yield substantively similar results to Figure 3.

# 6 HOW MANY PEOPLE INJECT DRUGS IN KOHTLA-JARVE, ESTONIA?

According to the European Drug Report 2023, from 2015-2021 Estonia had the highest per capita prevalence of People Who Inject Drugs (PWID) in Europe. There is also evidence of high HIV prevalence [Degenhardt et al., 2017] and drug overdose death rates (related to the introduction of Fentanyl) among PWID in Estonia during this time period [Uusküla et al., 2020]. To lower the prevalence of HIV among PWID in Estonia, syringe exchange programs were launched in 1997 [Wu et al., 2017]. Estimating the size of the PWID population sheds light on the magnitude of this public health crisis and the necessary scope of potential policy solutions.

Wu et al. [2017] use data from an RDS sample conducted in 2012 to estimate the number of PWID in the Kohtla-Jarve region of Estonia. They compare a series of models including the standard multiplier method [Fearon et al., 2017], successive sampling [Johnston et al., 2010], and a network-based approach [Crawford et al., 2018b]. This RDS sample began with 6 seeds and includes 600 participants from the Kohtla-Jarve region. The data on each member of the study includes their arrival time, degree, recruiter identity, and allotted coupons. We use the IIE approach of Section 4, estimating the population size to be $\widetilde{N} = 795$ and the average wait time to be $1/\widetilde{\lambda}_n = 1/0.23$. This is within the intervals implied by previous estimates [Wu et al., 2017].

These data further include an indicator of whether the participant is using antiretroviral therapy (ART) for HIV. We use this covariate and the RDS sample to construct a data-realistic simulation study to showcase how a hypothetical network-based covariate could assist in estimating population size. A simple change to the study could have asked each person to share their ART status with their social connections in the PWID population (to hopefully increase screening for HIV and uptake of ART). The auxiliary information to be collected from each RDS participant is then a measurement of how many people have shared their ART status with them since the beginning of the study. Letting $\mathbf{x}_{ART} \in \{0,1\}^n$ be the indicator of ART status, the responses to this question, $\mathbf{Q} = (q_1, q_2, \ldots, q_n)$, could follow a Poisson model similar to the one described in Section 5,

$$q_i \sim \text{Poisson}\left(\left[\left\{A^S \cdot \text{lt}(\mathbf{1}_n \mathbf{1}_n^\top)\right\} \mathbf{x}_{ART}\right]_i\right).$$

The simulation proceeds as follows: we first select an $A^S$ that is compatible with the observed RDS. Treating this $A^S$ as ground truth, we set $N = 1105$ (this is the most likely population size that could have generated that $A^S$). Finally, we set $\lambda = \widetilde{\lambda}_n = 0.23$ as was estimated without auxiliary information. We incorporate the auxiliary information in $\mathbf{Q}$

Table 2: Population Estimation MAD

| Algorithm | MAD | Std. |
|----------:|----:|----:|
| MLE | 219.1 | 9.3 |
| IIE w/ Info | 181.3 | 6.8 |

This table displays the MAD of population size estimates for the case study of Section 6.

to improve our estimation of $N$ as outlined in Section 4.2. Table 2 compares $\widetilde{N}^r$ and $\widehat{N}$, and we see that when such auxiliary information is available, leveraging it improves population size estimation (by approximately 20%).

# 7 CONCLUSION

RDS provides access to populations often excluded from scientific discourse. Although this sampling process presents a variety of inferential problems, it also contains valuable information on the social network connecting study participants. This paper expands on the existing literature with new mechanisms for improving estimation of the study participant arrival rate, complete subgraph, and population size. The first accounts for the the bias of the MLE using concepts from indirect inference, and the second proposes a mechanism for including auxiliary information. Both methods combine to achieve cutting edge performance.

Although modeling arrival time data as independent and exponential is natural, loosening this assumption would allow for more realistic dependencies in the RDS model. For example, future work could consider wait times that depend on recruiter covariates (accounting for recruiter preferences). Additionally, the inferential advantage of including auxiliary information in the estimation procedure depends on the quality of this data. Future research could focus on optimizing auxiliary information collection for inferential targets such as population size and degree distribution. Lastly, accurate recovery of the sample subgraph is essential for tasks beyond population size estimation, such as running randomized experiments and measuring the efficacy of interventions on the RDS sample.

# 8 ACKNOWLEDGEMENTS

The authors would like to thank the reviewers for numerous helpful comments. In conducting this research, Alexander Volfovsky was partially supported by the National Science Foundation (NSF) Faculty Early Career Development Award DMS-2046880 and NSF award DMS-2230074. Justin Weltz was partially supported by NSF award DMS-2046880 as well. Eric Laber acknowledges support from the NSF award DMS-CDS&E-MSS-2346292 and the National Institutes of Health award R01DA056407.

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

# Supplementary Material

**Justin Weltz**[1]  **Eric Laber**[1,3]  **Alexander Volfovsky**[1,2]

[1]Department of Statistical Science, Duke University, Durham, North Carolina, USA
[2]Department of Computer Science, Duke University, Durham, North Carolina, USA
[3]Department of Biostatistics and Bioinformatics, Duke University, Durham, North Carolina, USA

## A  ADDITIONAL SIMULATION RESULTS FOR SECTION 3.3

This section contains simulation results that are referenced in Section 3.3 of the main text. It continues the empirical evaluation of the IIE for the two model parameters in Equation (1): $A^S$, the subgraph between study participants, and $\lambda$, the study paricipant arrival rate. Table 1 in Section 3.3 of the main text shows that $\widetilde{A}_n^S$, the IIE of $A^S$, has a higher true positive rate than $\widehat{A}_n^S$, the MLE of $A^S$, across all simulation settings.

We first evaluate the error rates of $\widetilde{A}_n^S$ and $\widehat{A}_n^S$ in more detail. Table 3 reports the true negative rates (TNR) of $\widehat{A}_n^S$ and $\widetilde{A}_n^S$ over a range of graph densities and population sizes. It shows that there is no discernible difference between the TNR of the IIE and MLE in these settings. Therefore, the higher true positive rates of $\widetilde{A}_n^S$ depicted in Table 1 do not come at the expense of overall accuracy.

We also compare the performance of the IIE and the MLE for $\lambda$ in terms of MSE. Table 4 shows that the IIE, $\widetilde{\lambda}_n$, is considerably more accurate than the MLE, $\widehat{\lambda}_n$, over a range of graph sizes and densities. We observe that $\widetilde{\lambda}_n$ has an MSE that is less than 50% of the MSE of $\widehat{\lambda}_n$ across all settings. Additionally, the difference in MSE between $\widetilde{\lambda}_n$ and $\widehat{\lambda}_n$ is slightly higher with larger population sizes, which correspond to lower sample proportions (since the sample size is held fixed at $n = 100$), and higher average degrees.

Table 3: True Negative Rates of $\widehat{A}_n^S$ and $\widetilde{A}_n^S$ (%)

|  |  | MLE | | IIE | |
|---|---|---|---|---|---|
| Pop. | Deg | Average | Std. | Average | Std. |
| 1000 | 5 | 99.61 | 0.01 | 99.61 | 0.01 |
| 1000 | 10 | 99.08 | 0.01 | 99.08 | 0.01 |
| 1000 | 15 | 98.61 | 0.02 | 98.61 | 0.02 |
| 5000 | 5 | 99.92 | 0.00 | 99.92 | 0.00 |
| 5000 | 10 | 99.83 | 0.01 | 99.82 | 0.01 |
| 5000 | 15 | 99.72 | 0.01 | 99.72 | 0.01 |
| 10000 | 5 | 99.96 | 0.00 | 99.96 | 0.00 |
| 10000 | 10 | 99.90 | 0.00 | 99.90 | 0.00 |
| 10000 | 15 | 99.87 | 0.01 | 99.87 | 0.00 |

These are the true negative rates of $\widehat{A}_n^S$ and $\widetilde{A}_n^S$ for a series of population sizes (Pop.) and average degrees (Deg.). The standard deviations reported quantify the Monte Carlo error associated with these estimates based on 100 simulations.

Table 4: MSE of $\widehat{\lambda}_n$ and $\widetilde{\lambda}_n$

| | | IIE | | MLE | |
|---|---|---|---|---|---|
| Pop. | Deg. | Mean | Sd | Mean | Sd |
| 1000 | 5 | 0.09 | 0.02 | 0.21 | 0.02 |
| 1000 | 10 | 0.11 | 0.02 | 0.28 | 0.03 |
| 1000 | 15 | 0.09 | 0.02 | 0.24 | 0.03 |
| 5000 | 5 | 0.11 | 0.02 | 0.25 | 0.02 |
| 5000 | 10 | 0.13 | 0.03 | 0.36 | 0.04 |
| 5000 | 15 | 0.10 | 0.02 | 0.27 | 0.03 |
| 10000 | 5 | 0.10 | 0.02 | 0.28 | 0.03 |
| 10000 | 10 | 0.12 | 0.03 | 0.32 | 0.04 |
| 10000 | 15 | 0.09 | 0.02 | 0.29 | 0.03 |

These are the MSEs of the $\lambda$ estimators for a series of population sizes (Pop.) and average degrees (Deg.). The standard deviations reported quantify the Monte Carlo error associated with these estimates over 100 simulations.

## B   STOCHASTIC BLOCK MODEL ANALYSIS

This section provides the details of Simulation 2 from Section 5 in the main text. In this experiment, we test the performance of our population size estimators in a more complex graph model setting. The Erdos-Renyi model we employ assumes that edges between members of the population form with the same probability, $p$. However, individuals may be more likely to form connections with one group of people than another. Consider the following generative model for the population graph, $G = (V, E)$. The hidden population is divided into two groups, $V_A \subseteq V$ and $V_B = V \setminus V_A$ with sizes $N_A = |V_A|$ and $N_B = |V_B|$. The probability of an edge between members of the same group is $p_{\text{in}}$, and the probability of a connection between members of different groups is $p_{\text{out}}$. For constant $c \in [0, 1]$, we set $p_{\text{out}} = cp_{\text{in}}$ so that $p_{\text{in}} \geq p_{\text{out}}$. This is an example of a stochastic block model (SMB), which is used throughout network analysis [Holland et al., 1983, Lee and Wilkinson, 2019, Khabbazian et al., 2017]. We let $p^* = \mathbb{P}(\{i, j\} \in E)$, where nodes $i$ and $j$ are drawn uniformly at random from $V$. Defining $E_{\text{out}}$ and $E_{\text{in}}$ as the set of edges between and within groups respectively, we derive an expression for $p^*$ in terms of $p_{\text{out}}, c, N_A$, and $N_B$,

$$
\begin{aligned}
p^* &= \mathbb{P}(\{i, j\} \in E) \\
&= \mathbb{P}(\{i, j\} \in E_{\text{out}}) * p_{\text{out}} + \mathbb{P}(\{i, j\} \in E_{\text{in}}) * p_{\text{in}} \\
&= \frac{2N_A N_B}{(N_A + N_B)(N_A + N_B - 1)} p_{\text{out}} \\
&\quad + \frac{N_A(N_A - 1) + N_B(N_B - 1)}{(N_A + N_B)(N_A + N_B - 1)} p_{\text{in}}.
\end{aligned}
$$

Because $cp_{\text{out}} = p_{\text{in}}$,

$$
p^* = \frac{2cN_A N_B + N_A(N_A - 1) + N_B(N_B - 1)}{(N_A + N_B)(N_A + N_B - 1)} p_{\text{out}}.
$$

We use this expression to set the overall edge prevalence in the simulations below, making $N = 5000$ and $Np^* = 10$.

We assess the performance of the population size estimators in the SBM setting. Assume that we observe the group membership of each study participant, and define $i_A, i_B$ as the number of individuals in groups $A$ and $B$ respectively that are recruited before participant $i + 1$. Labelling $N_i = \mathbb{I}(i \in V_A)(N_A - i_A) + \{1 - \mathbb{I}(i \in V_A)\}(N_B - i_B)$,

$$
\mathbb{P}(d_i^u = d) = \sum_{j=0}^{d} \binom{N_i}{j} p_{\text{in}}^j (1 - p_{\text{in}})^{N_i - j} \binom{(N - i) - N_i}{d - j} p_{\text{out}}^{d-j} (1 - p_{\text{out}})^{(N-i) - N_i - (d-j)}. \tag{6}
$$

We define the following estimators based on Equation 6,

$$\left\{\widehat{p}_{\text{in}}, \widehat{p}_{\text{out}}, \widehat{N}\right\} = \arg \max_{\check{p}_{\text{in}}, \check{p}_{\text{out}}, \tilde{N}} \log \mathcal{L}(\check{N}, \check{p}_{\text{in}}, \check{p}_{\text{out}} | \widehat{A}_n^S) + R_{\text{out}}(\check{p}_{\text{out}}) + R_{\text{in}}(\check{p}_{\text{in}}),$$

$$\left\{\widetilde{p}_{\text{in}}, \widetilde{p}_{\text{out}}, \widetilde{N}\right\} = \arg \max_{\check{p}_{\text{in}}, \check{p}_{\text{out}}, \tilde{N}} \log \mathcal{L}(\check{N}, \check{p}_{\text{in}}, \check{p}_{\text{out}} | \widetilde{A}_n^S) + R_{\text{out}}(\check{p}_{\text{out}}) + R_{\text{in}}(\check{p}_{\text{in}}), \tag{7}$$

$$\left\{\widetilde{p}_{\text{in}}^r, \widetilde{p}_{\text{out}}^r, \widetilde{N}^r\right\} = \arg \max_{\check{p}_{\text{in}}, \check{p}_{\text{out}}, \tilde{N}} \log \mathcal{L}(\check{N}, \check{p}_{\text{in}}, \check{p}_{\text{out}} | \widetilde{A}_n^{S,r}) + R_{\text{out}}(\check{p}_{\text{out}}) + R_{\text{in}}(\check{p}_{\text{in}}),$$

where $R_{\text{out}}(\check{p}_{\text{out}}) = \log \text{Beta}(\check{p}_{\text{out}}; a_{\text{out}}, b_{\text{out}})$ and $R_{\text{in}}(\check{p}_{\text{in}}) = \log \text{Beta}(\check{p}_{\text{in}}; a_{\text{in}}, b_{\text{in}})$.

In our simulation, $c = 0.3$, $N_A/N = 0.6$, and auxiliary information is as specified in Simulation 1 of Section 5. We assess the performance of correctly and incorrectly specified estimators, recognizing that the correctly specified estimators require observing the group label of study participants (but not of their unobserved neighbors). The correctly specified estimators are listed in Equation 7, and the incorrectly specified estimators assume the Erdos-Renyi model. For every correctly specified estimator, $e^{R_{\text{out}}(\check{p}_{\text{out}})} = \text{Beta}(\check{p}_{\text{out}}; a_{\text{out}}, b_{\text{out}})$ and $e^{R_{\text{in}}(\check{p}_{\text{in}})} = \text{Beta}(\check{p}_{\text{in}}; a_{\text{in}}, b_{\text{in}})$ are centered at $p_{\text{in}}$ and $p_{\text{out}}$ respectively with $a_{\text{out}}, a_{\text{in}} = 10$. For the misspecified estimators, we center $e^{R(\check{p})} = \text{Beta}(\check{p}; a, b)$ at $p_{\text{in}}$ and set $a = 10$. Figure 4 indicates that the estimators based on the correctly specified likelihood perform an order of magnitude better than the estimators assuming the Erdos-Renyi model. Additionally, the relationships between $\widehat{N}$, $\widetilde{N}$, and $\widetilde{N}^r$ mirror the results in the Erdos-Renyi setting (Figure 3). Lastly, both the correctly and incorrectly specified network-based estimators outperform the successive sampling estimator with $N_{\text{max}}/N = 3$.

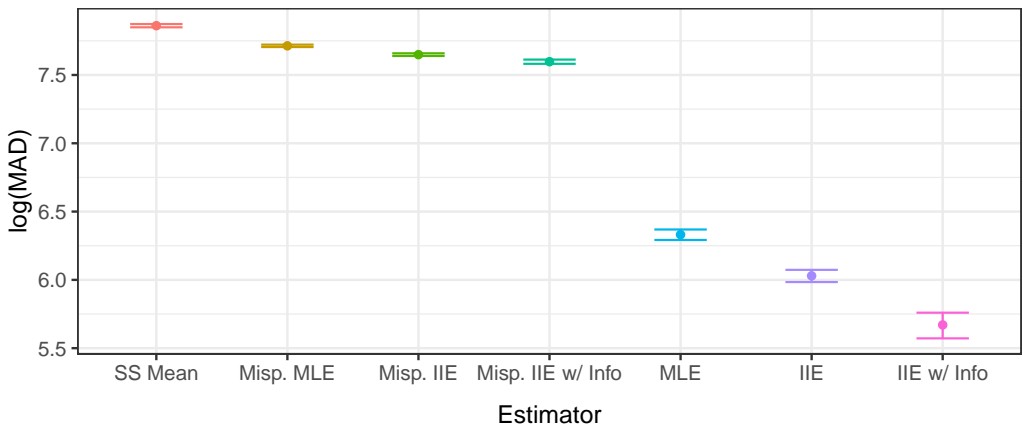

Figure 4: This figure compares the performance of a successive sampling estimator (mean of the posterior distribution), $\widehat{N}$, $\widetilde{N}$, and $\widetilde{N}^r$ in the SBM setting. The estimators proceeded by "Misp." incorrectly assume the Erdos-Renyi model. The figure includes 90% Monte Carlo confidence intervals for each estimator.

We assess the sensitivity of incorrectly specified population size estimators to the SBM setting more extensively in Table 5 (auxiliary information is also as specified in Simulation 1 of Section 5). To illustrate the effect of misspecification, we vary $N_A/N$ and $c$. As $N_A/N \to 1$ (or 0) or $c \to 1$, the Erdos-Renyi model becomes a better approximation of the truth. As $N_A/N \to 0.5$ and $c \to 0$, there is more heterogeneity in the graph edge probabilities, and the approximation becomes worse. We can see this pattern in Table 5. The first line of the table, $N_A/N = 1$ and $c = 1$, shows the MAD of our population size estimators under the Erdos-Renyi model for comparison. When $c = 0.3$ and $N_A/N = 0.5$, the error of the estimators is highest, and when $c = 0.9$ and $N_A/N = 0.75$, it is lowest; i.e., when $c = 0.3$ and the groups are evenly split, the estimators demonstrate a $150\% - 300\%$ increase in MAD over the estimators in the Erdos-Renyi model setting, while when $c = 0.9$ and $N_A/N = 0.75$, the increase is only $19\% - 34\%$. The rest of Table 5 illustrates a continuous spectrum between these two extremes. Lastly, as mentioned in Section 5 of the main text, the incorrectly specified IIEs still perform better than their MLE counterparts.

## C   LATENT SPACE SENSITIVITY RESULTS

In this section, we present a sensitivity analysis referenced in Section 5. Instead of drawing the hidden population graph from an Erdos-Renyi distribution, we generate it from a latent space inner product model in 2-dimensions. In this context,

Table 5: Population Size Estimation MAD with Incorrect Strong Regularization Information in the SBM Setting

| $N_A/N$ | c | MLE | IIE | IIE w/ Info |
|---|---|---|---|---|
| 1.0 | 1.0 | 861.6 | 560.9 | 459.4 |
| 0.50 | 0.3 | 2318.7 | 2071.8 | 2021.4 |
| 0.50 | 0.6 | 1711.1 | 1425.1 | 1344.8 |
| 0.50 | 0.9 | 1116.1 | 755.6 | 638.7 |
| 0.75 | 0.3 | 1784.1 | 1519.8 | 1443.4 |
| 0.75 | 0.6 | 1502.3 | 1178.5 | 1079.9 |
| 0.75 | 0.9 | 1039.5 | 672.6 | 614.2 |

This table displays the Mean Absolute Deviation (MAD) of the population estimators over a series of $N_A/N$ and $c$ values. We use strong regularization information centered at $p_{\text{in}}$ ($a = 10$ in the standard $\log \text{Beta}(\breve{p}; a, b)$ regularizer term) to mimic ignorance of the two block structure. These results are averaged over simulations with Monte Carlo standard deviations below 25.

each member of the hidden population, $i \in V$, has an unobserved "position," $x_i \in \mathbb{R}^2$, in latent space. The probability of an edge between individuals $i, j \in V$ is dictated by the inner product between $x_i$ and $x_j$,

$$\mathbb{P}\left(\{i, j\} \in E\right) = \frac{e^{\phi_0 + \phi_1 x_i^\top x_j}}{1 + e^{\phi_0 + \phi_1 x_i^\top x_j}},$$

where $\phi_0, \phi_1 \in \mathbb{R}$. In this simulation setting, we set the population size equal to 5000. For each $i \in V$, we draw $x_i$ independently,

$$x_i \sim \text{Normal}\left\{(0, 0)^\top, \begin{pmatrix} 0.01 & 0 \\ 0 & 0.01 \end{pmatrix}\right\},$$

and set $\phi_0 = -6.21$ and $\phi_1 = 1$. This results in an expected overall degree of about 10. Under these parameters, the edge probabilities are approximately $1.5 \times 10^{-3}$ to $2.7 \times 10^{-3}$. The regularization term is set to $\log \text{Beta}(\breve{p}; a, b)$, where $\text{Beta}(\breve{p}; a, b)$ is centered at an approximation of the overall edge density with $a = 0.1$ (weak regularization). Figure 5 indicates that the maximum likelihood estimator, $\widehat{N}$, based on the Erdos-Renyi assumption still has a lower maximum absolute deviation (MAD) than the successive sampling estimator with $N_{\max}/N = 5$. Additionally, the indirect inference estimator, $\widetilde{N}$, and the indirect inference estimator with auxiliary information, $\widetilde{N}^r$ (using the same information as Simulation 1 of Section 5), outperform $\widehat{N}$. This provides evidence that our estimation methods are still advantageous in misspecified settings.

# D    SIMULATION RESULTS UNDER STRONG REGULARIZATION FOR SECTION 5

In this section, we present the results of a simulation under strong (correctly specified) regularization. As described in Section 4.1 of the main text, we use a regularized MLE approach to estimate population size to avoid specifying informative priors that are difficult to justify scientifically. The regularization function, $R(\breve{p}) = \log \text{Beta}(\breve{p}; a, b)$, incorporates information on edge prevalence, $p$ — where $\text{Beta}(\breve{p}; a, b)$ is a Beta distribution that is centered at $p$ with a variance that is inversely proportional to $a$. Here, we use the same setup as Simulation 1 but vary the hyperparameters in the regularizer.

In Figure 3 of Section 5 in the main text, we compare the MAD of $\widehat{N}$ (MLE), $\widetilde{N}$ (IIE), and $\widetilde{N}^r$ (IIE with auxiliary information) with $a = 0.1$, and a series of Successive Sampling (SS) estimators. We observe that $\widetilde{N}^r$ improves on $\widetilde{N}$, and both are more accurate than $\widehat{N}$. Additionally, the performances of the SS estimators are highly dependent on their prior. In Figures 6a and 6b, we show the log(MAD) of $\widehat{N}, \widetilde{N}$, and $\widetilde{N}^r$ with $a = 1$ and $a = 10$ respectively. The relationships between estimators $\widehat{N}, \widetilde{N}$ and $\widetilde{N}^r$ mirror Figure 3. Encouragingly, the MAD of our population size estimators decreases significantly as $a$ increases, and, with strong regularization information, $\widehat{N}, \widetilde{N}$ and $\widetilde{N}^r$ are consistently more accurate than the SS estimators.

# E    IIE AND SUCCESSIVE SAMPLING ALGORITHM DETAILS

In this section, we present the details of Algorithms 1 and 2 (Algorithm 1 is described in Section 3 of the main text).

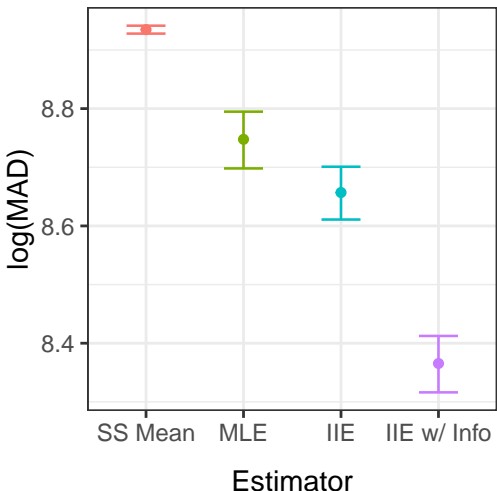

Figure 5: This figure compares the performance of a successive sampling estimator (mean of the posterior distribution), $\widehat{N}, \widetilde{N}$, and $\widetilde{N}^r$ when the Erdos-Renyi assumption is violated by drawing the population graph from a latent space model. It includes 90% Monte Carlo confidence intervals for each estimator.

Both algorithms construct the IIE by finding the parameters under which the expected value of a calibration statistic is equal to the observed value, where the calibration statistic is set equal to the MLE of $\lambda$. In the simulations of Section 5 in the main text, we use $K = 9$ grid values centered at $\widehat{\lambda}_n$, the MLE for the observed data. Specifically, we set $\lambda^k = \widehat{\lambda}_n - (k-4) \times 0.1$ for $k \in \{1, 2, \ldots, 9\}$. The set of candidate parameters are $\left\{ \lambda^k, A_\lambda^S(\mathbf{Y}, \lambda^k) \right\}_{k=1}^9$ and $\left\{ \lambda^k, A_\lambda^{S,r}(\mathbf{Y}^r, \lambda^k) \right\}_{k=1}^9$ for Algorithms 1 and 2 respectively. For each candidate parameter, we approximate the expected value (setting $J = 25$) of the MLE of $\lambda$, labeling this quantity $\widehat{\lambda}_n^k$. The IIE is the set of parameters under which $\widehat{\lambda}_n^k$ is closest to $\widehat{\lambda}_n$.

As described in Section 4.2 of the main text, Algorithm 2 augments Algorithm 1 with auxiliary information. We note that this implies the MLE is taken with respect to different likelihoods in Algorithms 1 and 2. Defining $\beta \in \mathbb{R}^p$ for $p \in \mathbb{N}$ as the parameter that indexes the distribution of $\mathbf{Q}$, the MLE referenced in Algorithm 2 is

$$\left\{ \widehat{A}_n^S, \widehat{\lambda}_n, \widehat{\beta}_n \right\} = \arg \max_{A^S \in \mathcal{A}, \lambda \in \mathbb{R}^+, \beta \in \mathbb{R}^p} \mathcal{L}_n(\mathbf{Y}, \mathbf{Q}|A^S, \lambda, \beta).$$

Algorithms 1 and 2 take about 24 hours to run with a sample size of 100. They are implemented in the code included in the Supplementary Material.

Lastly, we use the SSPSE package [Handcock et al., 2023] under a "flat" prior setting to construct the SS estimators analyzed in Figures 3 and 6.

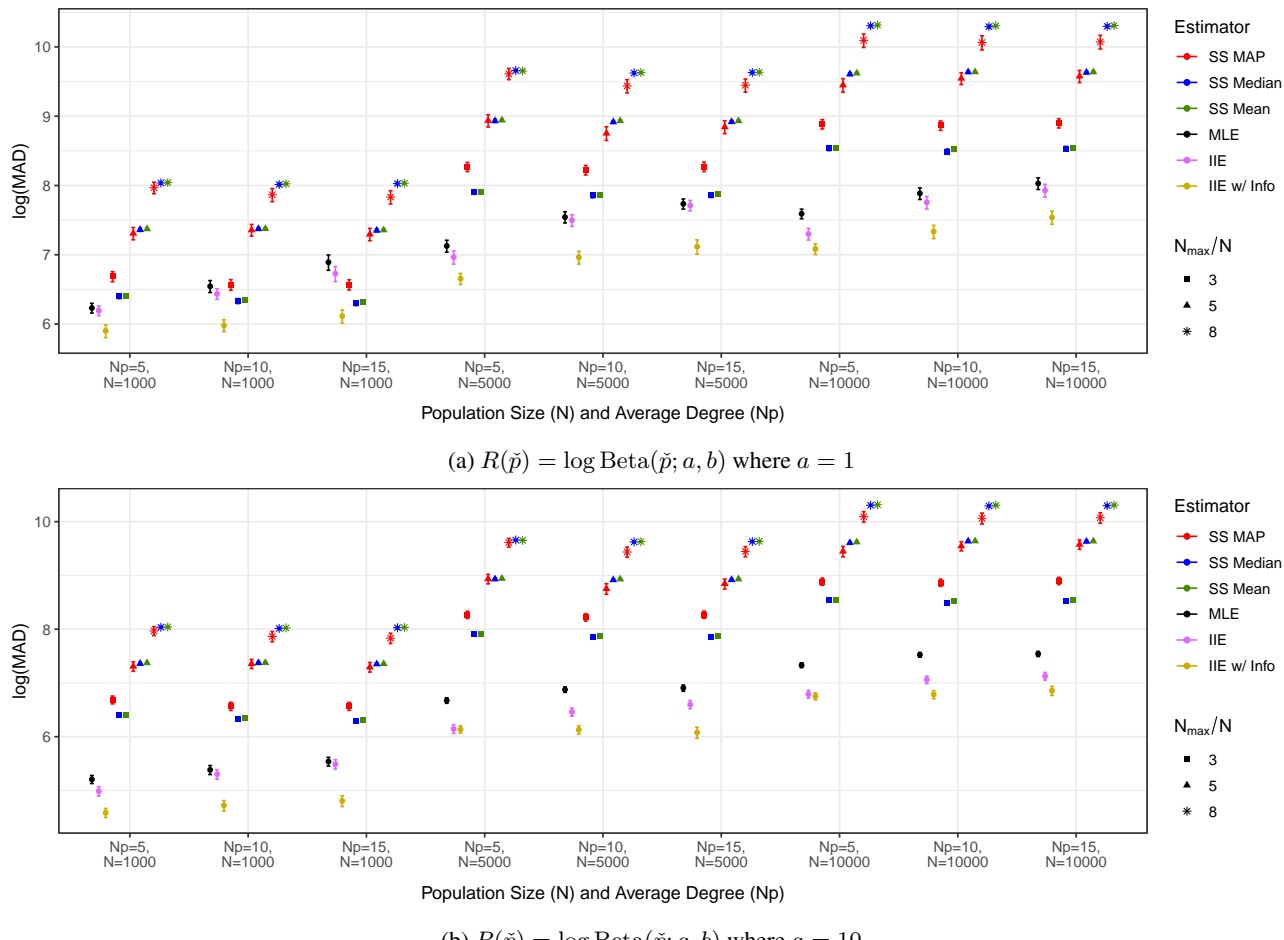

(a) $R(\check{p}) = \log \mathrm{Beta}(\check{p}; a, b)$ where $a = 1$

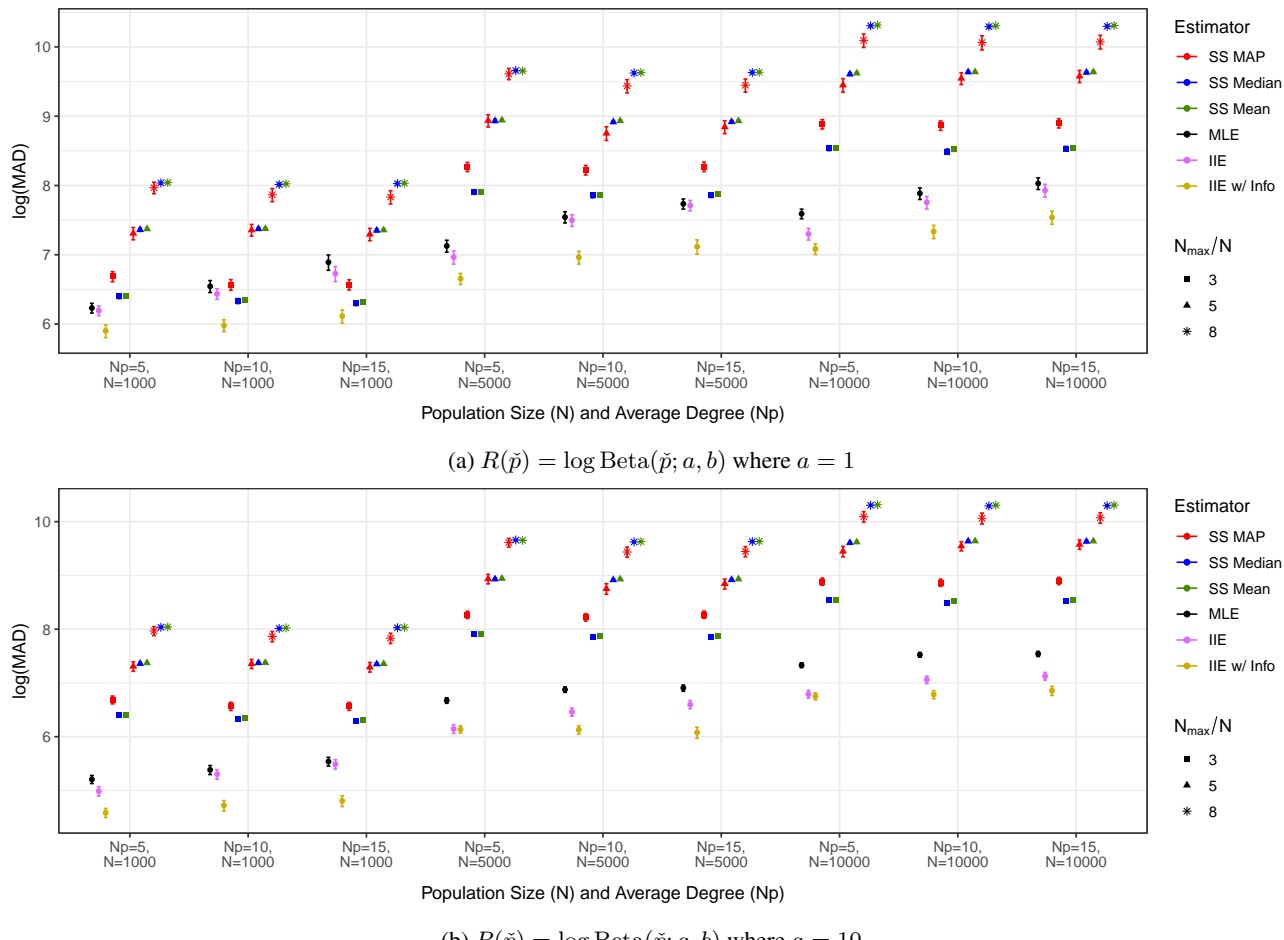

(b) $R(\check{p}) = \log \mathrm{Beta}(\check{p}; a, b)$ where $a = 10$

Figure 6: This figure compares the performance of $\widehat{N}, \widetilde{N}$, and $\widetilde{N}^r$ under strong regularization information over a series of population sizes, $N$, and average degrees, $Np$, with 90% Monte Carlo confidence intervals.

---

**Algorithm 2:** The Indirect Inference Estimator with Auxiliary Information

---

We want to find the estimator,

$$\widetilde{\lambda}_n^r \in \arg \min_{\lambda \in \mathbb{R}^+} \left| \mathbb{E}_{\mathbf{Z} \sim P_{A_\lambda^{S,r}(\mathbf{Y}^r, \lambda), \lambda}} \left\{ \lambda^\dagger(\mathbf{Z}^r) \right\} - \lambda^\dagger(\mathbf{Y}^r) \right| ;$$

Generate a grid of $\lambda^k$ values, $k \in \{1, 2, .., K\}$ ;
**for** $k$ *in* $\{1, 2, .., K\}$ **do**
  **for** $j$ *in* $\{1, 2, .., J\}$ **do**
    Find $A_{k,j}^S = \max_{A^S, \beta} \mathcal{L}(A^S, \beta | \lambda^k, \mathbf{Y}^r)$, where $\beta$ indexes the distribution of $\mathbf{Q}$;
    Simulate wait time vector $w^{k,j}$ from the model defined by parameters $A_{k,j}^S, \lambda^k$;
    Find $\widehat{\lambda}_n^{k,j}, \widehat{A}_n^{S,k,j}$ by maximizing the likelihood given the generated data $\mathbf{Z}_{k,j}^r = (w^{k,j}, G^R, \mathbf{d}, \mathbf{Q})$;
  **end**
  Save vector $(\lambda^k, A_k^S, \beta^k, \widehat{\lambda}_n^k = \frac{\sum_{j=1}^J \widehat{\lambda}_n^{k,j}}{n})$;
**end**
Calculate $k^* = \arg \min_k \left| \widehat{\lambda}_n^k - \lambda_r^\dagger(\mathbf{Y}^r) \right|$;
Our estimator is then

$$\left( \widetilde{\lambda}_n^r, \widetilde{A}_n^{S,r}, \widetilde{\beta}^r \right) = \left( \lambda^{k^*}, A_{k^*}^S, \beta^{k^*} \right)$$