# OpenReview forum: "Hidden Population Estimation with Indirect Inference and Auxiliary Information"
_auai.org/UAI/2024/Conference — UAI 2024 spotlight_

### Official Review · Reviewer_i3gn · 2024-02-29

**Q2-1 Originality-Novelty:** 3
**Q2-2 Correctness-Technical Quality:** 3
**Q2-5 Clarity Of Writing:** 3

**Q10 Ethical Concerns:**

No.

**Q1 Summary And Contributions:**

Estimating the size of a population is a difficult task when the population of interest is defined by illegal or stigmatized behaviors. Respondent Driven Sampling (RDS) is a common method used in this context to collect data, but existing methods that estimate the underlying structure of the population are biased. In this work, the authors propose methods that improve on the accuracy of estimating the total population from an RDS sample by using indirect inference and auxiliary information.

**Q2-3 Extent To Which Claims Are Supported By Evidence:**

4: Excellent: all claims are supported by very convincing evidence (in the form of comprehensive experimental evaluation, rigorous mathematical proofs, detailed (pseudo-)code, precise references, well-motivated and realistic assumptions) and the authors deliver what they promise.

**Q2-4 Reproducibility:**

4: Excellent: key resources (e.g. proofs, code, data) are available and key details (e.g. proof sketches, experimental setup) are comprehensively described for competent researchers to confidently and easily reproduce the main results.

**Q3 Main Strengths:**

I believe the main strengths of this paper are as follows:
1. The scope and discussion of this paper are impressively comprehensive. The authors describe the RDS model, current state of the art methods, why current methods are insufficient, two proposed improvements over the current state of the art, and synthetic and non-synthetic experimental evaluations of their methods. The discussions cover all essential aspects of the problem.

2. This paper includes rigorous mathematical discussions and comprehensive code that allow scientists in the field to easily reproduce their synthetic and non-synthetic results. Transparency is important in science communication, and this paper is an excellent example of it.

3. The organization of the paper makes key concepts easy to follow; there is a natural flow in the discussion of the paper.

**Q4 Main Weakness:**

I believe the main weaknesses of this paper are as follows:
1. My main critique of this paper is that I wish there was more discussion and justification of key assumptions aside from citing previous papers in the field. Specifically, why are Assumptions 1 and 2 things that are reasonable in your problem setup, and what do these assumptions encode at a high level? I understand that these assumptions may be widely accepted in the field of RDS sampling, but I wish there was more discussion about what they mean at a high level and why they are reasonable.

2. The synthetic simulations all use the Erdos-Renyi model of graph generation. Are there other models in the field that are important and/or widely used that warrant further discussion? Do the authors’ methods still work under more complex models of graph generation?

3. The authors do not have a non-synthetic experiment using real-world data that evaluates their method for approximating population size with auxiliary information.

**Q5 Detailed Comments To The Authors:**

Could you please put a border around the legend in Figure 2 to increase readability?

I might have missed this, but how might one generate the initial values lambda^k in Algorithm 1?

I understand that the initial submission has space constraints, but could the authors please include a discussion about potential future work?

**Q9 Complying With Reviewing Instructions:**

Yes

---

> ### Author Rebuttal · Authors · 2024-04-04
>
> Thank you for your thoughtful review. It is encouraging to hear that the paper was clear, reproducible, and comprehensive.
>
> Response to Weakness 1:  Assumption 1 is common when studying arrival data. It implies Markovian dynamics and leads to a closed form likelihood for the respondent driven sampling (RDS) process (Equation 1). It is possible to relax this assumption, e.g., by considering dependence in arrival times. This will lead to changes in the likelihood but does not preclude the analytic approach we propose — i.e., by making all of the required changes, we can perform estimation under any wait time distribution. This would involve computing the likelihood by evaluating the distribution of the minimum of the susceptible edge times at each recruitment event. We will add a discussion of the limitation of this assumption and of possible solutions to the final version of this manuscript.
>
> We included Assumption 2 for completeness. It is a technical assumption about the existence of an Edgeworth expansion. As we are dealing with standard distributions, including this assumption explicitly may have been unnecessary. In the absence of the network nature of our problem, Assumption 2 is easily verified for exponential wait times. This is a reasonable assumption for more complex models that are built on top of the exponential likelihood. We will include a more extensive discussion of this assumption in the final version of this manuscript.
>
> Response to Weakness 2: We want to note that we do include derivations and simulation studies where the underlying truth is not an Erdos-Renyi (ER) model. In Section 4, we derive the likelihood for population size estimation under a stochastic block model (SBM). The SBM is a generalization of the ER model that allows for edge probabilities to depend on community structure. We report experimental results under the SBM in Section 5 and Appendix B. Additionally, we present sensitivity analyses in Section 5, supported by further exposition in Appendices C and D, that demonstrate the practical implications of assuming an ER structure when the true graph model is more complicated. These results show that our estimators continue to perform better than competitors under a range of SBM and latent space models despite misspecification.
>
> Response to Weakness 3: Unfortunately, we do not have real-world data where the ground truth population size is known (this could potentially be collected in a stylized experiment on a college campus for example). Nonetheless, we wanted to demonstrate the strength of our approach over existing methods using a real-world dataset. Consequently, we included a synthetic example that is based on real RDS data. In Section 6, we preserve the reported auxiliary information and recruitment graph while selecting a “true” $A^S$ that is compatible with the observed data. Table 2 shows that our estimator outperforms the maximum likelihood estimator in this experiment.
>
> Response to Detailed Comments: We will make sure to put a border around the Figure 2 legend, describe how the $\lambda^k$ values are generated in the main text (we set the $\lambda^k$ values to be a grid centered at the maximum likelihood estimate, $\hat{\lambda}_n$, as described in Appendix E), and include a discussion of future work. Natural extensions of our paper include non-exponential wait times (potentially varying with recruiter covariates), optimizing auxiliary information collection for graph and population size inference, and treatment effect estimation over the recruited sample that incorporates interference along the underlying network.

---

### Official Review · Reviewer_ai4R · 2024-03-20

**Q2-1 Originality-Novelty:** 3
**Q2-2 Correctness-Technical Quality:** 3
**Q2-5 Clarity Of Writing:** 3

**Q1 Summary And Contributions:**

This paper proposes a novel approach leveraging indirect inference and auxiliary participant information to improve the accuracy of population size estimates. They address biases inherent in existing estimation methods by employing a two-stage procedure that enhances the estimation of the social network's underlying structure and, subsequently, the hidden population size.

**Q2-3 Extent To Which Claims Are Supported By Evidence:**

3: Good: the main claims are supported by convincing evidence (in the form of adequate experimental evaluation, proofs, (pseudo-)code, references, assumptions).

**Q2-4 Reproducibility:**

4: Excellent: key resources (e.g. proofs, code, data) are available and key details (e.g. proof sketches, experimental setup) are comprehensively described for competent researchers to confidently and easily reproduce the main results.

**Q3 Main Strengths:**

1. The considered problem is interesting and of practical value.

2. The approach is validated through simulations and a real-world application, showcasing its effectiveness and practical utility.

**Q4 Main Weakness:**

Perhaps more real-world examples could be helpful for readers not familiar with the topic to understand the theory better. However, this may not really be a main weakness since I'm not familiar with the topic.

**Q5 Detailed Comments To The Authors:**

Please refer to the main weakness.

**Q9 Complying With Reviewing Instructions:**

Yes

---

> ### Author Rebuttal · Authors · 2024-04-04
>
> Thank you for your thoughtful review. We will add more discussion of settings where respondent driven sampling (RDS) is used in practice and can be improved by the proposed methodology. For example, RDS data on people who inject drugs (PWID) in St. Petersburg, Russia has been used in multiple studies (Crawford, 2016; Crawford et al., 2018; Heimer and White, 2010). Studying PWID in this setting is epidemiologically important because HIV incidence and prevalence among this population is high. However, PWID in St. Petersburg cannot be sampled through conventional methods because this population will not readily self-identify (drug possession in Russia carries severe legal penalties). Consequently, RDS is leveraged because it is easier to engender trust if study participants recruit their own social contacts. RDS has been similarly employed in numerous settings to study hidden populations at risk of HIV and other infectious diseases (Remera et al., 2024; Mapingure et al., 2024; Alinaghi et al., 2024; Barry et al., 2024).
>
> Additionally, Tyldum and Johnston (2014) have published a comprehensive book surveying RDS studies of migrant workers. They argue that RDS is a natural sampling mechanism for collecting information on these hard-to-reach populations. Reliable data are necessary in this context as “more and more people cross national and international borders, and labor markets become increasingly reliant on migrant labor" (Tyldum and Johnston, 2014). RDS has also been used for hard-to-reach populations such as street children (Johnston et al., 2010), the unhoused population (Bernard et al., 2018), and ethnic minorities (Mullo et al., 2020).
>
> Lastly, RDS is essential when the relevant population is highly stigmatized (Stahlman et al., 2016; Arayasirikul et al., 2015; Magno et al., 2020). For example, Shahmanesh et al. (2009) studies suicidal behavior among female sex workers in Goa, India. This research is imperative for effective public health policy because sex workers in this setting are subject to high levels of HIV prevalence and violence.

---

### Official Review · Reviewer_8zhX · 2024-03-23

**Q2-1 Originality-Novelty:** 4
**Q2-2 Correctness-Technical Quality:** 3
**Q2-5 Clarity Of Writing:** 4

**Q1 Summary And Contributions:**

The authors study the problem of population estimation using respondent driven sampling. This study proposes leveraging auxiliary participant informaiton and using indirect inference to improve the estimation of population size. The authors proposed method reduces bias and improves the estimation of population size. Experiments on the real world dataset support authors proposed method.

**Q2-3 Extent To Which Claims Are Supported By Evidence:**

3: Good: the main claims are supported by convincing evidence (in the form of adequate experimental evaluation, proofs, (pseudo-)code, references, assumptions).

**Q2-4 Reproducibility:**

3: Good: key resources (e.g. proofs, code, data) are available and key details (e.g. proofs, experimental setup) are sufficiently well-described for competent researchers to confidently reproduce the main results.

**Q3 Main Strengths:**

1. This paper is extremely relevant and i found it a very complelling problem to be solved.
2. Incorporating the common auxiliary information into the estimation of social network improves the population estimates.
3. The estimation of participant arrival rate and population size could be cruicial in several domains with real world impact on an average individual.

**Q4 Main Weakness:**

1. Can the authors discuss other domains where this method can have impact.
2. I understand that IIE reduces bias, however, can there be any ethical concerns if information about the participants is shared with / without their consent. I would like to understand how the authors would like to address privacy concerns of auxiliary information.

**Q5 Detailed Comments To The Authors:**

please see above

**Q9 Complying With Reviewing Instructions:**

Yes

---

> ### Author Rebuttal · Authors · 2024-04-04
>
> Thank you for your thoughtful review. It is encouraging to hear that the paper is compelling and relevant.
>
> Response to Weakness 1: We will add more discussion of settings where respondent driven sampling (RDS) is used in practice and can be improved by the proposed methodology. For example, RDS data on people who inject drugs (PWID) in St. Petersburg, Russia has been used in multiple studies (Crawford, 2016; Crawford et al., 2018; Heimer and White, 2010). Studying PWID in this setting is epidemiologically important because HIV incidence and prevalence among this population is high. However, PWID in St. Petersburg cannot be sampled through conventional methods because this population will not readily self-identify (drug possession in Russia carries severe legal penalties). Consequently, RDS is leveraged because it is easier to engender trust if study participants recruit their own social contacts. RDS has been similarly employed in numerous settings to study hidden populations at risk of HIV and other infectious diseases (Remera et al., 2024; Mapingure et al., 2024; Alinaghi et al., 2024; Barry et al., 2024).
>
>  Additionally, Tyldum and Johnston (2014) have published a comprehensive book surveying RDS studies of migrant workers. They argue that RDS is a natural sampling mechanism for collecting information on these hard-to-reach populations. Reliable data are necessary in this context as “more and more people cross national and international borders, and labor markets become increasingly reliant on migrant labor" (Tyldum and Johnston, 2014). RDS has also been used for hard-to-reach populations such as street children (Johnston et al., 2010), the unhoused population (Bernard et al., 2018), and ethnic minorities (Mullo et al., 2020).
>
> Lastly, RDS is essential when the relevant population is highly stigmatized (Stahlman et al., 2016; Arayasirikul et al., 2015; Magno et al., 2020). For example, Shahmanesh et al. (2009) studies suicidal behavior among female sex workers in Goa, India. This research is imperative for effective public health policy because sex workers in this setting are subject to high levels of HIV prevalence and violence.
>
> Response to Weakness 2: In the context of hidden populations, privacy concerns are of the upmost importance. RDS is designed to promote anonymity and trust, and we would extend these considerations to the collection of auxiliary information. To this end, the example described in Section 4.2 and the case study of Section 6 only use auxiliary information inherent to the study participants and the referral chain. Importantly, they do not require sensitive information to be shared about participants. We will add a discussion of these considerations in the final version of this manuscript.

---

### Official Review · Reviewer_8ikX · 2024-03-23

**Q2-1 Originality-Novelty:** 3
**Q2-2 Correctness-Technical Quality:** 3
**Q2-5 Clarity Of Writing:** 3

**Q1 Summary And Contributions:**

This paper proposes an indirect inference estimator for estimating hidden populations. By leveraging auxiliary information and indirect inference, biases in estimation are reduced, enhancing accuracy in estimating population size and related parameters.

**Q2-3 Extent To Which Claims Are Supported By Evidence:**

3: Good: the main claims are supported by convincing evidence (in the form of adequate experimental evaluation, proofs, (pseudo-)code, references, assumptions).

**Q2-4 Reproducibility:**

3: Good: key resources (e.g. proofs, code, data) are available and key details (e.g. proofs, experimental setup) are sufficiently well-described for competent researchers to confidently reproduce the main results.

**Q3 Main Strengths:**

Strengths:
- Debias existing estimators of the underlying social network in an RDS via indirect inference.
- The asymptotic analyses are conducted.

**Q4 Main Weakness:**

Weaknesses:
- This paper proposes a debised estimator to estimate the hidden population size. It could be better to provide the application/motivation to estimate the population size.
- Why current method have bias, and where does the bias come from?

**Q5 Detailed Comments To The Authors:**

See the weakness above.

**Q9 Complying With Reviewing Instructions:**

Yes

---

> ### Author Rebuttal · Authors · 2024-04-04
>
> Thank you for your thoughtful review.
>
> Response to Weakness in Motivation: Estimating the hidden population size is often imperative for assessing the magnitude of a public health crisis as illustrated by our case study. As mentioned in the manuscript, estimating the number of people who inject drugs (PWID) sheds light on the magnitude of this public health crisis and the necessary scope of potential policy solutions. In the final version of this manuscript, we will discuss the demographic, public health, and epidemiological motivations for population size estimation explicitly in the introduction.
>
> Response to Weakness in Bias Explanation: In Section 2.1, we discuss the issues with the current method. We argue that the bias of the maximum likelihood estimator (MLE) is related to the natural bias of the MLE for the rate parameter of the exponential distribution. When combined with loose graphical constraints, this leads to the problem illustrated in Figure 2. We will include a further explanation of the MLE’s bias in the final version of this manuscript.

---

### Official Review · Reviewer_dpB7 · 2024-03-31

**Q2-1 Originality-Novelty:** 3
**Q2-2 Correctness-Technical Quality:** 3
**Q2-5 Clarity Of Writing:** 4

**Q10 Ethical Concerns:**

No.

**Q1 Summary And Contributions:**

This paper addresses the issues with inference using data collected using Respondent Driven Sampling (RDS). The paper proposes to use indirect inference and auxiliary participant information to tackle the issues and improve the estimations of participant arrival rate, complete sample subgraph and hidden population size with RDS data. Simulation studies are conducted to demonstrate the performance and superiority of the proposed method in comparison with state-of-art RDS inference methods and a case study is also conducted to use the proposed method to estimated the hidden population size of People Who Inject Drugs (PWID) in a region of Estonia.

**Q2-3 Extent To Which Claims Are Supported By Evidence:**

3: Good: the main claims are supported by convincing evidence (in the form of adequate experimental evaluation, proofs, (pseudo-)code, references, assumptions).

**Q2-4 Reproducibility:**

3: Good: key resources (e.g. proofs, code, data) are available and key details (e.g. proofs, experimental setup) are sufficiently well-described for competent researchers to confidently reproduce the main results.

**Q3 Main Strengths:**

1. The is paper is very well written, with easy-to-understand language without losing technical rigorous, and a clear logic flow.
2. The proposed method and perspective/information used to tackle the RDS issues look reasonable and well justified.
3. The experiment results have provided convincing support of the efficacy of the propose method.

**Q4 Main Weakness:**

1. Some assumptions of the proposed approach and their practical implications should be discussed in more detail. o
2. Some more elaboration on the technical details is needed.
3. The evaluation and/or demonstration of significance can be further strengthened.
Please see more details on the weakness in the next section.

**Q5 Detailed Comments To The Authors:**

1. Assumptions and their practical implications
It would be useful to provide more discussion on the assumptions, e.g. the Erdo-Renyi graph assumption, the chance for them to hold in practice, and the practical implications if they are violated in practice.

Is there any requirement on $\mid V^S\mid$ compared to $N$? Does it has to be large enough for the results to hold, or does it just need to be $\le N$?

2. Technical details
In Table 1, the std values of IIE are much bigger than the std values of MLE. Any explanation on this?

It could be just my eyes, but in Figure 3, the color used for IIE is very close to the color for SS MAP  so that it is difficult to distinguish the results of the two methods.

3. Evaluation/significance demonstration
Overall the evaluation part is well designed, but in terms of case studies,

(a)  the fact that the result obtained by IIE "is within the intervals implied by previous estimates" has not provided a strong evidence of the superiority of IIE, given that the paper is aimed at "improvement";

(b) it would be better to show more case studies of IIE in terms of improvement in estimating participant arrival time and complete participant subgraph to demonstrate the significance/impact of the work in the paper.

**Q9 Complying With Reviewing Instructions:**

Yes

---

> ### Author Rebuttal · Authors · 2024-04-04
>
> Thank you for your thoughtful review. We are happy the paper was easy-to-understand, well justified, and convincingly supported by experimental results.
>
> Response to Detailed Comment 1: While the Erdos-Renyi (ER) graph model is standard in the literature (Crawford et al., 2018; Frank and Snijders, 1994; Killworth et al., 1998), our setup for estimating the population size does not rely on this assumption. In Section 4, we derive the likelihood for population size estimation under a stochastic block model (SBM). The SBM is a generalization of the ER model that allows for edge probabilities to depend on community structure. We report experimental results under the SBM in Section 5 and Appendix B. Additionally, we present sensitivity analyses in Section 5, supported by further exposition in Appendices C and D, that demonstrate the practical implications of assuming an ER structure when the true graph model is more complicated. These results show that our estimators continue to achieve lower maximum absolute deviation (MAD) than competitors under a range of SBM and latent space models despite misspecification.
>
> Additionally, in the theoretical results we simply require $|V^S|$ to grow with $N$.
>
> Response to Detailed Comment 2: Table 1 reports Monte Carlo standard errors over 100 simulations — we only report these errors to demonstrate that even at 100 simulations it is clear that the indirect inference estimator (IIE) performs better than the maximum likelihood estimator (MLE) at recovering the true network between study participants (through the true positive rate and other rates reported in Appendix A).
>
> Additionally, we will fix the color scheme in Figure 3.
>
> Response to Detailed Comment 3: It is encouraging to hear that the simulation experiments were well designed. For the case study, we simply report that the indirect inference estimator "is within the intervals implied by previous estimates" to demonstrate that our results are reasonable. As there is no ground truth, we cannot argue that our method is an improvement. However, in the last paragraph of Section 6 we construct a synthetic experiment grounded in the case study: we select a true $A^S$ that is compatible with the observed data while preserving the reported auxiliary information and recruitment graph. Table 2 shows that our estimator outperforms the MLE in this setting. We will include a clearer explanation of the case study in the final version of this manuscript.

---

### Meta-Review · Area_Chair_yMTG · 2024-04-17

This paper tackles challenges associated with inference using data gathered via Respondent Driven Sampling (RDS). It suggests employing indirect inference and auxiliary participant information to address these challenges and enhance estimations of participant arrival rate, complete sample subgraph, and hidden population size with RDS data. Simulation studies and a case study are conducted to showcase the performance and superiority of the proposed method compared to state-of-the-art RDS inference methods. With unanimous positive feedback on the novelty and technical quality of this paper from all reviewers, I recommend accepting this paper.